# Survival of In-Hospital Cardiac Arrest in COVID-19 Infected Patients

**DOI:** 10.3390/healthcare9101315

**Published:** 2021-10-01

**Authors:** Mohammad Aldabagh, Sneha Wagle, Marie Cesa, Arlene Yu, Muhammad Farooq, Ythan Goldberg

**Affiliations:** Montefiore Medical Center, New York, NY 10467, USA; swagle@montefiore.org (S.W.); marie.cesa2019@gmail.com (M.C.); rleneyu@gmail.com (A.Y.); mfarooq@montefiore.org (M.F.); ygoldber@montefiore.org (Y.G.)

**Keywords:** COVID, pandemic, cardiac arrest

## Abstract

Background: There are limited data regarding the outcome of in-hospital cardiopulmonary resuscitation (CPR) in COVID-19 patients. In this study, we compared the outcomes of in-hospital cardiac arrests (IHCA) before and at the peak of the COVID-19 pandemic at Montefiore Medical Center in the Bronx, New York, United States. We also identified the most common comorbidities associated with poor outcomes in our community. Methods: This was a multi-site, single-center, retrospective, observational study. Inclusion criteria for COVID patients were all confirmed positive cases who had in-hospital cardiac arrest (IHCA) between 1 March 2020 and 30 June 2020. The non-COVID cohort included all cardiac arrest cases who had IHCA in 2019. We excluded all out-of-hospital cardiac arrest (OHCA). We compared actual survival to that predicted by the GO-FAR score, a validated prediction model for determining survival following IHCA. Results: There were 334 cases in 2019 compared to 450 cases during the specified period in 2020. Patients who initially survived cardiac arrest but then had their code statuses changed to do not resuscitate (DNR) were excluded. Groups were similar in terms of sex distribution, and both had an average age of about 66 years. Seventy percent of COVID patients were of Black or Hispanic ethnicity. A shockable rhythm was present in 7% of COVID patients and 17% of non-COVID patients (*p* < 0.05). COVID patients had higher BMI (30.7 vs. 28.4, *p* < 0.05), higher prevalence of diabetes mellitus (58% vs. 38%, *p* < 0.05), and lower incidence of coronary artery disease (22% vs. 35%, *p* < 0.05). Both groups had almost similar predicted average survival rates based on the GO-FAR score, but only 1.5% of COVID patients survived to discharge compared to 7% of non-COVID patients (*p* < 0.05). Conclusion: The rate of survival to hospital discharge in COVID-19 patients who suffer IHCA is worse than in non-COVID patients, and lower than that predicted by the GO-FAR score. This finding may help inform our patient population about risk factors associated with high mortality in COVID-19 infection, as well as educate hospitalized patients and healthcare proxies in the setting of code status designation.

## 1. Introduction

Since the initial detection of COVID-19 cases in the United States in February 2020, more than 40 million cases have been confirmed, with over 656,000 deaths to date [1]. Recent studies estimate that 12–19% of COVID-19 positive patients may require hospitalization, with 3–6% becoming critically ill [2]. Among critically ill patients, hypoxemic respiratory failure secondary to acute respiratory distress syndrome (ARDS), myocardial injury, ventricular arrhythmias, and shock are common and predispose them to cardiac arrest [3].

Patients with COVID-19 requiring mechanical ventilation have poor survival rates [3,4]. Furthermore, survival after cardiac arrest is likely substantially lower. The incidence of out-of-hospital cardiac arrest (OHCA) and death during the COVID-19 pandemic significantly increased compared with the same period the previous year [5]. However, there are limited data regarding the outcome of in-patient cardiac arrest (IHCA) in COVID-19 patients [6,7].

The first study to report outcomes of in-hospital cardiac arrest (IHCA) in COVID-19 was from Wuhan, China, where the survival rate of patients with severe COVID-19 pneumonia and in-hospital cardiac arrest was very poor, with a return of spontaneous circulation (ROSC) rate of 13.2% and a 30-day survival rate of 2.9% [6].

In this study, we compared the baseline characteristics and survival rates of COVID and non-COVID patients who experienced IHCA at our institution, with the aim of gaining insight into risk factors associated with poor outcomes and to highlight the importance of goals of care discussions for critically ill COVID-19 patients.

## 2. Methods

The electronic medical record (EMR) system from three affiliated hospitals in the Bronx, NY, United States, was used for patient selection and data collection. All patients with COVID-19 confirmed by either nasopharyngeal swab or SARS-CoV-2 who were aged 18 and above, had IHCA and underwent cardiac resuscitation (CPR) between 15 March and 30 June 2020 were included in the COVID cohort. The non-COVID cohort consisted of all patients who had in-hospital cardiac arrests and underwent cardiac resuscitation in the period 1 January to 31 December 2019. We used a longer time period for non-COVID patients to increase the size of the comparison group and to reach better matching between the two groups. Patients with OHCA or who arrived to the hospital with ongoing cardiac resuscitation were excluded. Internal transfers within the three hospitals were considered a single hospitalization. Those who were transferred outside the institution or who had cardiac arrest prior to transfer to our institution were excluded. Patients whose code statuses were changed to do not resuscitate (DNR) after initial survival of cardiac arrest were also excluded. This study was approved by the local institutional review board.

All data were obtained from EMRs and cardiac arrest documentation. Patients’ data included age, gender, race, body mass index (BMI), code status as either full code or (DNR), and comorbidities such as hypertension, diabetes mellitus (DM), coronary artery disease, and chronic kidney disease. Other data included location of cardiac arrest, initial electrocardiogram (ECG) rhythm during the cardiac arrest, and length of hospital stay.

The GO-FAR score (Good Outcome Following Attempted Resuscitation) is a clinically validated risk score that utilizes pre-arrest variables to predict the probability of survival to hospital discharge following IHCA [8]. These variables include age, neurologic status on admission, organ status on admission (kidney, liver, cardiac or respiratory failure), and cardiac versus non-cardiac diagnosis on admission. Survival rate categories as defined by the GO-FAR score are shown in Table 1.

Descriptive statistics with simple graphic analysis were performed. Numerical variables are reported as mean, median, or range as appropriate. Categorical variables are reported as the percentage of patients in each category. Proportions were compared using the Wald test. A *p*-value <0.05 was considered statistically significant. STATA-16 was used for analysis.

## 3. Results

There were 334 patients who had IHCA in 2019. Of the 6500 COVID-positive patients who were admitted between 15 March and 30 June 2020, we found 450 patients who underwent cardiac resuscitation efforts for IHCA. Age and sex distribution were similar in both groups (Table 2). Seventy-two percent of COVID patients were Black and Hispanic, while sixty-eight percent of non-COVID patients were Black and Hispanic. Seventy percent of COVID patients had shockable rhythms during cardiac arrest compared to seventeen percent of non-COVID patients (*p* < 0.05). COVID patients had higher BMIs (30.7 vs. 28.4, *p* < 0.05) and a higher prevalence of diabetes mellitus (58% vs. 38%, *p* < 0.05). Coronary artery disease was more common in non-COVID patients (35% vs. 22%, *p* < 0.05). Despite both groups having almost similar predicted average-to-low survival rates based on the GO-FAR score, only 7 (1.5%) COVID patients survived to discharge compared to 23 (7%) non- COVID patients (*p* < 0.05).

## 4. Discussion

A recent systematic review and a meta-analysis that included 40 studies in the period 1985–2018 showed that one-year survival after IHCA was 13.4% on average [9]. Our study demonstrates that the outcome of inpatient cardiac arrest for patients with COVID-19 was much lower, with a survival rate of 1.7%. This is comparable to the 2.9% survival rate reported in Wuhan, China [6]. The low survival rate after in-hospital cardiac arrest in COVID-19 patients raises the question of which factors may influence such poor outcomes [10].

### 4.1. Factors That Affect the Outcome of IHCA in Patients Infected with COVID-19

Multiple studies during the COIVD-19 pandemic have suggested that HTN, DM, and CAD are the most common comorbidities that affect the survival of COVID-19 patients, while coexisting infection, malignancy, immunodeficiency, and cerebrovascular disease are less common [11].

Numerous studies have found that the survival to hospital discharge is approximately 50% for patients who have a shockable rhythm during cardiac arrest. In contrast, the likelihood of survival from an initially non-shockable rhythm is two to three times lower (about 15–20%) [12]. In our study, we found that only 7% of COVID patients had a shockable rhythm compared to 17% of non-COVID patients (*p* < 0.05). The low survival rate among the COVID-19 patients may be attributed to the absence of shockable rhythms during the cardiac arrest event.

Age has been widely associated with reduced survival after cardiac arrest, especially for patients over the age of 60 [10]. Previous studies demonstrated that elderly patients with serious underlying medical conditions are not only at greater risk of COVID-19 infection but also at higher risk of COVID-related death [13]. This has remained consistent throughout the COVID pandemic and is further confirmed by our study. We found that all COVID-19 patients who survived to hospital discharge after a cardiac arrest were under age 65.

The average BMI in COVID patients was higher than in the non-COVID patients. This may contribute to the lower survival rate. The “obesity paradox” is a phenomenon that describes how obesity, as a risk factor for cardiac disease, can lead to improved mortality outcome in severe cardiac illness such as cardiac arrest. One study found high BMI to be associated with lower all-cause mortality in survivors of cardiac arrest [14]. However, obesity increases the risk of severe illness and death from COVID-19 [15].

Besides obesity, coronary artery disease (CAD) has also been associated with improved survival following cardiac arrest [16]. This has been attributed to ischemic preconditioning, as well as other mechanisms, such as the patients’ awareness of CAD symptoms and being on protective cardiac medications. The higher prevalence of CAD in non-COVID patients compared to COVID patients is also a likely contributor to the difference in survival in our study.

Diabetes mellitus (DM) was more common in our COVID cohort than in the non- COVID cohort. DM has been associated with reduced return to spontaneous rhythm and 24-h survival [17]. COVID patients with poorer blood glucose control were also found to have increased overall mortality [18].

Studies investigating the relationship between race and outcomes following in-hospital cardiac arrest found Black and Hispanic patients had lower rates of neurological recovery and survival compared to Caucasians [19]. Unfortunately, Blacks and Hispanics were most affected by COVID-19 infection, with the highest rates of hospitalization and death [20].

### 4.2. GO-FAR Score as an Estimate of Outcome after Cardiac Arrest

Despite similar predicted average survival rates based on the GO-FAR score, the actual survival for COVID patients was much lower for COVID patients than for non-COVID patients. One reason for this discrepancy may be that clinical features captured by the GO-FAR score may be incomplete or of insufficient magnitude with respect to COVID infection. Black and Hispanic patients, who represented a higher percentage of the COVID than non-COVID cohort, have been shown to have a lower survival following in-hospital cardiac arrest [20]. However, race and ethnicity are not included in the GO-FAR score. Another possibility is that the quality of CPR and other aspects of patient care in general could have suffered during the pandemic, leading to worse survival.

### 4.3. Effect of the Pandemic on the Health Care System, Quality of CPR and, Patient Care

CPR is a complex intervention that includes airway management, ventilation, chest compressions, drug therapy, and defibrillation. In the setting of COVID-19 infection, viral particles can remain in the air with a half-life of approximately 1 h, increasing the risk of transmission to resuscitation providers. Multiple studies have demonstrated a significant risk of viral transmission due to aerosol and droplet generation during CPR [21,22,23]. Due to this risk, hospital policies require providers to have full personal protective equipment (PPE) during CPR as per the American Heart Association guideline [24]. At the time of the early pandemic in the United States, efforts to preserve the source supply of PPE may have led to a reduction in the numbers of CPR responders [25].

In addition, during this unprecedented time in New York City, the number of critically ill patients exceeded the capacities of ICUs in most hospitals. Operating rooms, general medical floors, and hallways became sites to treat critically ill COVID-19 patients [26,27]. Pre- and post-cardiac arrest care may have been adversely impacted due to a combination of limited knowledge in treating COVID-19 infection, as well as the emotional burden on healthcare workers including feelings of fear, fatigue, and despair [28].

### 4.4. Study Limitations

This was a multi-hospital system study with a predominantly Black and Hispanic population with a high prevalence of significant comorbidities that explains the lower survival rate following IHCA in our community compared to the rate nationwide (7% vs. 13.4%). The two groups were chosen from different years and over different periods of time to avoid false negative results in the non-COVID group in 2020 and to achieve a better match between the two groups. Ten percent of COVID patients’ records regarding the rhythm during the cardiac arrest were missing due to the high volume of patients undergoing cardiac arrest with a limited number of providers. Undocumented details of cardiac arrest procedures included elements such as time of event, medications administered, ventilator setting at time of arrest, oxygen saturation and other vital signs. Additionally, as this study was carried out retrospectively, selection bias could not be completely avoided.

Due to the low number of cardiac arrest survivors at our institution, we could not statistically verify the relation between the risk factors and the outcome of cardiac arrest during the COVID pandemic.

## 5. Conclusions

The rate of survival to hospital discharge in COVID patients who suffer cardiac arrest is extremely low. The results from this study may inform providers, hospitalized patients, and their relatives when discussing designation of code status. This study also demonstrates the most common risk factors that affect the outcomes of inpatient cardiac arrest in COVID-19 patients. Further research with a larger and more diverse population is needed to better understand survival probability in specific patient subgroups.

## Figures and Tables

**Table 1 healthcare-09-01315-t001:** GO-FAR score.

GO-FAR Score	Risk Group	Survival to Discharge with Minimal Neurologic Disability
≥24	Very low survival	<1%
14 to 23	Low survival	1–3%
−5 to 13	Average survival	3–15%
−15 to −6	Above average survival	>15%
**GO-FAR Score Variables**	**Yes**	**No**
Neurologically intact or with minimal deficits at admission	−15	0
Major trauma Injury associated with shock or altered mental status during current admission	+10	0
Acute stroke Ischemic or hemorrhagic stroke during current admission	+8	0
Metastatic or hematologic cancer	+7	0
Septicemia Documented bloodstream infection with antibiotics not yet started or ongoing	+7	0
Medical noncardiac diagnosis on admission	+7	0
Hepatic insufficiency Total bilirubin >2 mg/dL or 34 µmol/L and AST >2× upper limit of normal, or cirrhosis	+6	0
Admit from skilled nursing facility	+6	0
Hypotension or hypoperfusion within 4 h prior to arrest SBP <90, MAP <60, pressors or inotropes other than dopamine ≤3 µmol/kg/min after volume expansion, or intra-aortic balloon pump	+5	0
Renal insufficiency or dialysis	+4	0
Respiratory insufficiency Any of the following: P/F ratio <300, PaO₂ <60, SaO₂ <90%; PaCO₂, ETCO₂, or transcutaneous CO₂ >50, spontaneous RR >40 or <5, or noninvasive or invasive ventilation	+4	0
Pneumonia Documented active pneumonia with antibiotics not yet started or still ongoing	+1	0

**Table 2 healthcare-09-01315-t002:** Characteristics of COVID patients compared to non-COVID patients.

Demographic Characteristics		Percentage (%)		*p*-Value
		Non-COVID (334)	COVID (450)	
Male		186 (55.7%)	271 (60.2%)	0.2
Age (years, mean + SD)		66.8 (15.5)	66.4 (13.1)	0.72
Race	White Black Spanish Asian Unavailable	49 (14.8%) 124 (37.3%) 92 (27.7%) 5 (1.5%) 62 (18.7%)	31 (6.9%) 173 (38.4%) 151 (33.6%) 13 (2.9%) 82 (18.2%)	<0.001 0.75 0.07 0.18 0.87
BMI (Kg/m^2^, mean ± SD)		28.4 (8.6)	30.7 (8.3)	<0.001
Location	General medical unit ED ICU	175 (52.4%) 31 (9.3%) 128 (38.3%)	236 (52.4%) 60 (13.3%) 154 (34.2%)	0.94 0.07 0.21
Rhythm	Non shockable Shockable No record	277 (82.9%) 56 (16.8%) 1 (0.3%)	370 (82.4%) 33 (7.3%) 46 (10.2%)	0.08 <0.001 <0.001
HTN		232 (69.5%)	340 (75.6%)	0.06
DM		128 (38.3%)	260 (57.8%)	<0.001
CAD		116 (34.7%)	97 (21.6%)	<0.001
Dialysis		81 (24.3%)	122 (27.1%)	0.36
GO-FAR score	Average survival rate Low survival rate	139 (41.6%) 93 (27.8%)	117 (26.1%) 123 (27.5%)	0.2 0.59
Length of stay (days, mean ± SD)		12.6 (15.1)	9.1 (11.7)	<0.001
Survivors to hospital discharge		23 (7%)	7 (1.5%)	<0.001

## Data Availability

Derived data supporting the findings of this study are available from the corresponding author on request.

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
