# Peer review of "Survival of In-Hospital Cardiac Arrest in COVID-19 Infected Patients"

_healthcare, 2021, doi:10.3390/healthcare9101315_

Round 1

Reviewer 1 Report

First of all, I would like to acknowledge the incredible work done by the caregivers in this sector (especially) during the pandemic and especially during its darkest hours. I would like to express my respect to the authors of this manuscript as well as to all the caregivers who worked with them.

I read with great interest the manuscript entitled "Survival of In-Hospital Cardiac Arrest in COVID-19 Infected Patients". The authors compared the prognosis of 2 historical cohorts, one of which consisted of 450 COVID+ patients who presented with IHCA which constitutes an impressive series of patients. However, I would have a few comments to make :

  • Methods section : could the authors please rephrase the sentence “ All confirmed... either by nasopharyngeal swab or SARS-CoV2”.
  • Methods section : could the authors to clearly state that the patients included in this study are patients for whom CPR was initiated ?
  • The variables necessary to calculate the GO-FAR score should appear in a separate table with the points attributed for each criteria
  • Table 2 : 10% of patients do not have record of the initial rhythm at the time of cardiac arrest. This missing data in a significant proportion of patients is a limitation of this study and should be acknowledged in the discussion
  • The authors should discuss the difference in length of stay between the 2 groups. Do they think that it shortened by the poor survival in the COVID group ?
  • I recognize that access to medical records is difficult: however, is it possible to determine if there were any pre-existing conditions and to describe them:

- hypotension/hypoperfusion

- pulmonary embolism (fibrinolysis?). Anticoagulant therapy in the COVID group ?

- arrythmia ?

- congestive heart failure ?

- septicemia ?

- what was the frequency of respiratory failure in the 2 groups?

- what proportion of patients were intubated at the time of cardiac arrest?

- what proportion of patients were receiving ECMO at the time of arrest and what proportion were receiving it during CPR?

- what was the duration of CPR before death/ROSC?

  • Table 3 does not seem to me to be easily understandable. It should be better described or removed.
  • Would it be possible to show in a table (apendix?) the characteristics of the surviving patients in the 2 groups and to make the comparison (and to discuss it)?
  • Page 4 : “In our study the average of BMI in COVID patients was lower than in non-COVID patients”. Table 2 says the opposite. Would it be possible for the authors to elaborate?

Author Response

thank you for feedback 

Reviewer 2 Report

Aldabagh et al. studied the outcome of in hospital cardiopulmonary resuscitation (CPR) in COVID-19 patients compared to non-COVID 19 patients. The idea is novel and authors provide  interesting results. some minor revisions should be made. 

1) there is an important limitation: non COVID patients were chosen from different years and periods. However, helathcare places, hospital admission and patient's features suffer from the particular pandemic period (see ESC Heart Fail2020 Oct 23;7(6):4182-4188. doi: 10.1002/ehf2.13043).  This aspect should be mentioned because it may further alter results. 

2)  define the abbreviations also in Tables

3) check carefully references' list (ref. n 7 is missing and 24 is listed after 25).

Author Response

1- We chose non covid patient from 2020 to avoid any false negative results in 2021 and also there were no enough non covid patients who underwent cardiac arrest in 2021 by the time of data gathering.  We will add more explaination and clarification to the manuscripts. 

2- Will add a table for abbreviation at the end of the paper. 

3- Will correct the reference list. 

Reviewer 3 Report

The manuscript: Survival of In-Hospital Cardiac Arrest in COVID-19 Infected Patients

(delete “.” At the end of title)

Abstract:

“at the height of the COVID-19 pandemic” Do you mean the level or degree?

“Inclusion criteria were all cardiac arrest cases who were resuscitated in 2019 and all confirmed COVID19 cases who had IHCA between March 1, 2020 and June 30, 2020.” Need to rewrite this sentence. It is quite confusing. All cases had included COVID-19 cases already. Was the data collection between March 1 and June 20?

“450 COVID+” you meant positive?

The conclusion: “The rate of survival to hospital discharge in COVID-19 patients who suffer IHCA is worse than in non-COVID patients, and lower than that predicted by the GO-FAR score. These findings may inform providers, hospitalized patients and healthcare proxies when discussing designation of code status.” According to the findings, patients with COVID had more comorbidities that may also affect their survival. I believe that the results may not only focus on the designation of code status but how to improve the survival of the COVID patients.

Introduction

This part is not enough to raise the necessity of conducting this study. Although the authors mentioned at the end to gaining insight into “factors” to help inform goals of care decisions….Was it one of the aims of the study? If so, the authors should add it in the abstract and in the introduction.

Results and discussion

In Table 3, Shockable rhythm had been reported with p<0.01. Do you mean <0.001? please check.

In Table 3, only shockable rhythm and +ve COVID-19 were significant to predict survival.  However, in the discussion section, age was not found to be associated with the survival. BMI was not entered into the regression model; obesity was also an insignificant result and so on. The authors must stick to the results and interpret accordingly. If the results did not show significant but the authors interpret them as significant factors, the regression was meaningless. I think the authors should revisit the data findings and interpret appropriately.

4.2 this paragraph GO-FAR score as an estimate of outcome after cardiac arrest seems like another part of limitation on the tool. What about the GO-FAR score findings between COVID and Non-COVID patients?

4.3 “Multiple studies have demonstrated a significant risk of viral transmission due to aerosol and droplet generation during CPR (21).” But only one citation. On the other hand, your results cannot show the relationship between CPR and PPE sources, and neither about the knowledge, experience to manage critically ill patients. The authors should be careful to handle the result data.

  1. Conclusion

The authors should conclude the findings and how these findings impact the current treatment and care. ie, the authors should address if the objectives have been achieved.

References

Need to check the list in accordance with the requirement of the journal. Eg #1 only the website is showed.

Round 2

Reviewer 3 Report

Dear authors,

The revised manuscript looks better but it's still not clear why factors are necessary to be identified in the introduction although the authors added in the study aims. 

It's not clear why another table was added under Table 1. There was no description regarding this table in the result. 

the authors mentioned that they intended to add some factors like age, obesity, BMI, etc in the discussion. The authors can compare their results regarding these factors and COVID-related mortality with previous studies. However, it's strange if your data did not show any findings on them but further described or emphasized. 

Author Response

Thank you for revieweing our paper, we appreciate your consideration and your feedback. 

1- we added the varibale of GO-FAR score under table-1 becuase anthoer reviewer asked for it. 

2- The factors that we discussed in our result like Age, BMI and shokable vs non shockable rhythm was notable in our study to be different between COVID and non- COVID patients and that why we said it might contribute to the low survival rate. We removed the regression analysis because the number of sruvival was very low to make it valid analysis but as a descriptive analysis we think it still valid to consider that those factors might affected the survival espically while many other studies confirmed that those factors badly affected the survival.